# Production, Characterization and Commercial Formulation of a Biosurfactant from *Candida tropicalis* UCP0996 and Its Application in Decontamination of Petroleum Pollutants

**Darne Germano Almeida** [1], **Rita de Cássia Freire Soares da Silva** [2,3], **Hugo Morais Meira** [2,3], **Pedro Pinto Ferreira Brasileiro** [3,4], **Elias José Silva** [4], **Juliana Moura Luna** [2,3], **Raquel Diniz Rufino** [2,3] and **Leonie Asfora Sarubbo** [1,2,3,]*

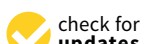



[1] Rede Nordeste de Biotecnologia (RENORBIO), Universidade Federal Rural de Pernambuco (UFRPE), Rua Dom Manuel de Medeiros, s/n-Dois Irmãos, Recife 52171-900, Pernambuco, Brazil; darnebio@yahoo.com.br

[2] Escola Icam Tech, Universidade Católica de Pernambuco (UNICAP), Rua do Príncipe, n. 526, Boa Vista, Recife 50050-900, Pernambuco, Brazil; rita.freire@iati.org.br (R.d.C.F.S.d.S.); hugo.meira@iati.org.br (H.M.M.); juliana.luna@unicap.br (J.M.L.); raquel.rufino@unicap.br (R.D.R.)

[3] Instituto Avançado de Tecnologia e Inovação (IATI), Rua Potyra, n. 31, Prado, Recife 50751-310, Pernambuco, Brazil; pedro.brasileiro@iati.org.br

[4] Departamento de Engenharia Química, Universidade Federal de Pernambuco (UFPE), Cidade Universitária, s/n, Recife 50740-540, Pernambuco, Brazil; eliasjose23@gmail.com

* Correspondence: leonie.sarubbo@unicap.br; Tel.: +55-81-2119-4084

**Abstract:** Contamination by oil and its derivatives causes serious damage to the environment, motivating the development of innovative technologies for the removal of these contaminants, such as the use of biosurfactants. In the present study, the biosurfactant from *Candida tropicalis* UCP0996 produced in the low cost-medium formulated with molasses, residual frying oil, and corn steep liquor, was characterized and its toxicity, formulation, and application in removal and biodegradation of oil were investigated. The surface tension of the medium was reduced to 30.4 mN/m, yielding 4.11 g/L of isolated biosurfactant after 120 h. Tests under extreme environmental conditions indicated the stability of the biosurfactant. Chemical characterization by thin layer chromatography (TLC), Fourier transform infrared spectroscopy (FTIR), nuclear magnetic resonance ($^1$H NMR), and gas chromatography and mass spectroscopy (CG-MS) revealed the glycolipidic nature of the biosurfactant. The isolated biosurfactant showed no toxicity against the microcrustacean *Artemia salina*, while the properties of the formulated biosurfactant remained stable during 120 days of storage. The biosurfactant removed 66.18% of motor oil adsorbed in marine stones and dispersed 70.95% of oil in seawater. The biosurfactant was also able to increase by 70% the degradation of motor oil by seawater indigenous microorganisms, showing great potential to be applied as a commercial additive in the bioremediation of oil spills.

**Keywords:** bioremediation; biosurfactant; *Candida tropicalis*; oil spills; surface tension

## 1. Introduction

The release of oil and by-products into the environment is a major cause of global pollution and has become a focus of great concern in both industrialized and developing countries, since oil pollution can have dramatic damaging effects on the environment and to the resident organisms. The main source of hydrocarbons in the oceans comes from routine ship washing operations, natural oil spills on the seabed, and accidents during oil exploration and transportation [1,2].

The need to remedy polluted areas has allowed the development of new technologies to treat contaminants, not only by chemical or physical methods, but also by biological techniques. Bioremediation allows the partial or total removal of contaminants through

biological activity [3]. For the success of bioremediation technologies, the use of microorganisms with metabolic abilities suitable for biodegradation and capable of transforming contaminants into less toxic substances is the most important requirement in oil spill bioremediation [4].

The *Candida tropicalis* yeast has been widely studied and considered a hydrocarbon degrading potent agent [5]. Studies have also reported that this species has the metabolic ability to produce biosurfactants under cultivation on water-immiscible substrates [6].

Biosurfactants are amphipathic molecules with hydrophobic and hydrophilic moieties that partition between phases with different polarities (oil/water and air/water), reducing the surface tension, and thus increasing the contact area of hydrocarbons, enhancing bioavailability, and biodegradation of such compounds [7]. The considerable interest in these compounds is related to their properties, as biodegradability, production from renewable substrates, low toxicity, biocompatibility, diversity for chemical structure and properties, effectiveness even at extreme conditions of temperature, pH, and salinity [2].

These features have intensified the studies on different applications for biosurfactants in the field of industrial bioremediation. The biosurfactant production has a considerable importance in the development of sustainable industrial processes through the use of renewable resources and "green" products [8,9].

The biosurfactant from the yeast *C. tropicalis* UCP 0996 described here was previously produced in bioreactors and applied in the removal of hydrophobic pollutants. The optimization of cultural conditions for the biosurfactant production using surface response methodology (SRM) was also described, showing promising results [10,11]. In the present study, the surfactant properties of the biosurfactant were determined. The biosurfactant was characterized and its toxicity was assessed. Finally, the biosurfactant was formulated and evaluated as a dispersing and bioremediation agent with a view to the industrial application of the biomolecule.

## 2. Materials and Methods

### 2.1. Materials

Waste frying oil was obtained from a local restaurant in the city of Recife-PE, Brazil, and stored according to the supplier's recommendations and used without any further processing. Corn steep liquor was obtained from Ingredion Brasil, Cabo de Santo Agostinho-PE, Brazil. Cane molasses was obtained from a local sugar mill in the city of Vitória de Santo Antão-PE, Brazil. Seawater was collected near the Thermoelectric TERMOPE, located in the city of Cabo de Santo Agostinho-PE, Brazil. Seawater samples were collected and stored in plastic bottles of 5 L.

### 2.2. Yeast Strain and Preparation of Inoculum

A strain of *Candida tropicalis* UCP0996 was provided from the culture collection of the Universidade Católica de Pernambuco, Recife city, Pernambuco, Brazil. The yeast was maintained at 5 °C on yeast mold agar (YMA) slants containing (*w/v*) yeast extract (0.3%), malt extract (0.3%), tryptone (0.5%), D-glucose (1.0%), and agar (5.0%), with pH 5.5. Inoculum was prepared by transferring cells to Erlenmeyer flasks containing 50 mL of yeast mold broth (YMB), which were kept under 200 rpm during 24 h at 28 °C.

### 2.3. Growth Curve and Biosurfactant Production

The biosurfactant was produced in a medium formulated with distilled water supplemented with 2.5% cane molasses, 2.5% residual frying oil and 2.5% corn steep liquor. The initial pH of the medium was adjusted to 5.5. Cultivation was carried out in 500 mL shake flasks under 200 rpm orbital shaking for 120 h at 28 °C. Samples were taken after 0, 2, 4, 6, 8, 16, 24, 32, 48, 72, 96, and 120 hours of fermentation to determine the growth profile (biomass), pH, surface tension, and biosurfactant yield.

### 2.4. Determination of Biosurfactant Properties

Surface tension was determined in the cell-free broth obtained after centrifugation at $10,000 \times g$ for 15 min with a Tensiometer (Sigma 700, KSV Instruments Ltd., Helsinki, Finland), using the Du Nouy ring method at 28 °C. The emulsification indexes for soybean, corn, and motor oils was determined according to Cooper and Goldenberg [12]. The ionic charge of the biosurfactant was determined as described by Rufino et al. [13].

### 2.5. Stability Studies

Stability studies were carried out in the cell-free broth (crude biosurfactant) obtained after centrifugation. The effect of pH was evaluated after adjustment of the samples pH to 2, 4, 6, 8, 10, and 12 with 6.0 M HCl or NaOH. The effect of NaCl (2.0%, 4.0%, 6.0%, 8.0%, 10.0%, and 12.0%) was determined after addition of the salt to the samples. The effect of heating time was evaluated in the samples kept at 90 °C during 10, 20, 40, 60, and 120 min, while the effect of temperature was carried out at for samples kept at 0, 5, 28, 70, 100, and 120 °C during 1 h and cooled to room temperature (28 °C). After preparation of the cell-free broth in the specific conditions of pH, NaCl, heating time, and temperature, samples were measured for surface tension and emulsification values. The assays were carried out in triplicate and did not vary more than 5%.

### 2.6. Determination of Cell Hydrophobicity

Cell hydrophobicity was determined by cell adhesion to hydrocarbons, following the method described by Rosenberg et al. [14]. Cells were washed twice in sterile deionized water and re-suspended in saline buffer (16.9 g/L $K_2HPO_4$; 7.3 g/L $KH_2PO_4$) to provide an optical density (OD) of 0.5–600 nm. One hundred μl of n-hexadecane were added to 2.0 mL of the cell suspension in a test tube (10 mm $\times$ 100 mm) and stirred in a vortex for 3 min. The contents were left to rest for 1 h for the separation of the aqueous and hexadecane phases. In the aqueous phase, OD was measured at 600 nm. Hydrophobicity was expressed as the percentage of adhesion to hexadecane and calculated as follows: $100 \times (1 - \text{aqueousphase OD/initial cell suspension OD})$. Three determinations were performed for each sample. Cells were considered very hydrophobic when rates exceeded 60% and poorly hydrophobic when the rate was under 10%.

### 2.7. Isolation of Biosurfactant

After cultivation (120 h), the biosurfactant was extracted from the cell-free broth following cell removal by centrifugation at $10,000 \times g$ for 15 min and filtered through Whatman nº.1 filter paper. An equal volume of $CHCl_3/CH_3OH$ (2:1) was placed with 50 mL of the cell-free broth in a separatory funnel at room temperature. The mixture was vigorously shaken for 15 min and allowed to set until phase separation. The organic phase was removed and the operation was repeated twice. The pooled product from organic phase was dried in an oven until complete evaporation of the solvent at 80 °C to a constant weight [15]. After extraction, the product was treated with a base and crystallized for maximum removal of impurities.

### 2.8. Determination of Surface Tension and Critical Micelle Concentration (CMC)

The surface tension was determined automatically by a Tensiometer (Sigma 700, KSV Instruments Ltd., Helsinki, Finland), using the Du Nouy ring method at 28 °C. The critical micelle concentration (CMC) was determined according to Almeida et al. [11].

### 2.9. Biosurfactant Characterization
#### 2.9.1. Thin-Layer Chromatography

A sample of 0.1 g of the isolated biosurfactant was dissolved in methanol and analyzed by thin layer chromatography (TLC) on silica gel plates. Chromatograms were developed with chloroform:methanol:acetic acid (65:15:2, $v/v$) and the detection was carried out by exposure to the Molish reagent for sugar detection, exposure to iodine vapors for lipid

stains and exposure to 1% ninhydrin solution for free amino groups. After spraying the reagents, plates were heated for 30–40 min at 110 °C to reveal the respective color and the retention factors (Rf) were calculated.

### 2.9.2. Nuclear Magnetic Resonance Spectroscopy

The extracted biosurfactant was re-dissolved in deuterated chloroform ($CDC_{l3}$) and the respective $^1$H NMR spectra were recorded at 25 °C using an Agilent 300 Mz spectrometer operating at 300.13 MHz. Chemical shifts ($\delta$) are given on the ppm scale relative to tetramethylsilane (TMS).

### 2.9.3. Fourier Transform Infrared Spectroscopy

The extracted biosurfactant was characterized by Fourier transform infrared spectroscopy (FTIR). The FTIR spectrum 400 Perkin Elmer with a resolution of 4 cm$^{-1}$ was collected from 400 to 4000 wavenumbers (cm$^{-1}$).

### 2.9.4. Gas Chromatography and Mass Spectroscopy

The hydrophobic moiety of the biosurfactant was analyzed on gas chromatograph-mass spectrometer system (Thermo Scientific Trace 1300—ISQ Single Quadrupole) equipped with a TGMS-5 column (30 m × 0.25 mm, 0.25 μm film thickness). Initial column temperature was 60 °C for 3 min, then ramped at 10 °C min$^{-1}$ to 300 °C and held for 15 min then a 1 μL sample was injected. Helium was used as carrier gas. The injector and detector temperatures were maintained at 300 and 280 °C, respectively.

### 2.10. Toxicity Against Artemia salina as Indicator

The toxicity of the biosurfactant was determined using the microcrustacean *Artemia salina*, as described by Meyer et al. [16]. Assays were conducted in penicillin tubes with 10 microcrustacean larvae containing seawater (5 mL) with solutions of the isolated biosurfactant at concentrations based on the CMC, until reach $LC_{50}$ (lowest concentration that kills 50% of tested brine shrimp) and on the cell-free broth (crude biosurfactant). Seawater was used as the control. Experiments were run in triplicate.

### 2.11. Biosurfactant Formulation

The cell-free broth (crude biosurfactant) obtained after centrifugation was subjected to different conservation methods: (a) addition of 0.2% potassium sorbate; (b) heating to 80 °C for 30 min (fluent vapor), followed by the addition of 0.2% potassium sorbate; and (c) sterilization at 121 °C for 30 min over three consecutive days (tyndallization). After applying each conservation method, the treated cell-free broth was stored at 28 °C and samples were withdrawn at 15, 30, 45, 90, and 120 days (long term stability study). After each storage time, the biosurfactant was subject to pH variations (5.0, 7.0, and 9.0), NaCl addition (1, 3, and 5% *w/v*) and heating (40 and 50 °C). Biosurfactant properties were checked by emulsification activity, surface tension determination, and motor oil spreading (dispersant) capacity in seawater.

### 2.12. Application of the Biosurfactant in Hydrophobic Contaminant Spreading

The oil spreading test was carried out by dropping 15 μL of motor oil slowly onto the surface of 40 mL of seawater layer contained in a Petri dish. Analyses were then conducted by the addition of 10 μL the formulated biosurfactant or the isolated biosurfactant solutions at $\frac{1}{2}\times$CMC, at CMC, at 2×CMC, and 5×CMC on the surface of the oil layer. The results were observed according to the halo formed and the dispersion was calculated based on the triplicate tests in relation to the diameter of the Petri dish [17].

### 2.13. Washing of Hydrophobic Compound Adsorbed to Porous Surface

Marine stones collected at Suape beach, Ipojuca city, Brazil, resulting from the fragmentation of coral reefs by the waves were used. The material was soaked in the contaminant

until completely covered, recording the volume spent. Then, each stone was carefully placed in a 100 mL beaker with the aid of tweezer, which were subjected to washing with the formulated biosurfactant and solutions of the isolated biosurfactant in concentrations of $\frac{1}{2}\times$CMC, CMC, 2×CMC, and 5×CMC. After the washing process, the removal percentage was calculated through the amount of oil remaining in the washed solid, by gravimetry, after extraction with hexane [18].

### 2.14. Swirling Bottle Test

A cylindrical open bottle of 1 L with an outlet valve at the bottom for taking samples was used in the dispersion experiment. Then, 200 mL samples of seawater were added in the bottle and 2 mL of oil was added to the water surface with a pipette. The formulated biosurfactant was dispensed in the center of the oil slick in the following proportions of biosurfactant to oil: 1:1, 1:2, 1:10, and 1:20 (*v*/*v*). The bottle was placed on an orbital shaker at 150 rpm at 28 °C for 10 min, followed by 1 to 2 min of rest to allow the larger drops to return to the surface. The samples were taken at 0, 5, and 10 min and extracted three times with hexane, as the biosurfactant is insoluble in hexane, to quantify the oil removed [19].

### 2.15. Bioremediation Test

For bioremediation tests, flasks with 100 mL of seawater, 1.0% motor oil, and solutions of the formulated biosurfactant and of the isolated biosurfactant ($\frac{1}{2}\times$CMC, CMC, 3×CMC, and 5×CMC) were incubated at 150 rpm at room temperature. Samples were analyzed after 1, 7, 14, 21, and 28 days for the number of microorganisms using the most probable number (MPN) [20].

### 2.16. Statistical Analysis

All presented experiments results were the average data of three independent replicates and were calculated using STATISTICA® program, version 10.0 (Statsoft Inc., Tulsa, OK, USA).

## 3. Results and Discussion

### 3.1. Growth Curve and Biosurfactant Production

The growth kinetics of the microorganism showed an exponential phase at 5–15 h of cultivation (Figure 1). Maximal biomass production (23.75 g/L) occurred after 120 h of fermentation. The lowest surface tension value also occurred in the stationary phase, with a reduction from 55 to 30.4 mN/m after 96 h, and the highest biosurfactant yield (4.11 g/L) occurred in the stationary growth phase. The pH of the medium decreased in the first 30 h and remained around 5.5 after 72 h of cultivation. El-Sheshtawy et al. [21] evaluated the biosurfactant production by *C. albicans* IMRU 3669. After 96 h of cultivation, the cell-free broth reduced the surface tension from 72 to 45 mN/m. In another study, Alwaely et al. [22] evaluated a biosurfactant produced by *Candida* spp. The results showed a biosurfactant yield of 9.0 g/L, and a reduction of the medium surface tension from 71 to 28 mN/m. Luna et al. [23], on the other hand, evaluated a biosurfactant produced by *C. bombicola*. They obtained a biosurfactant yield of 8.4 g/L and a surface tension of 30 mN/m after 120 h.

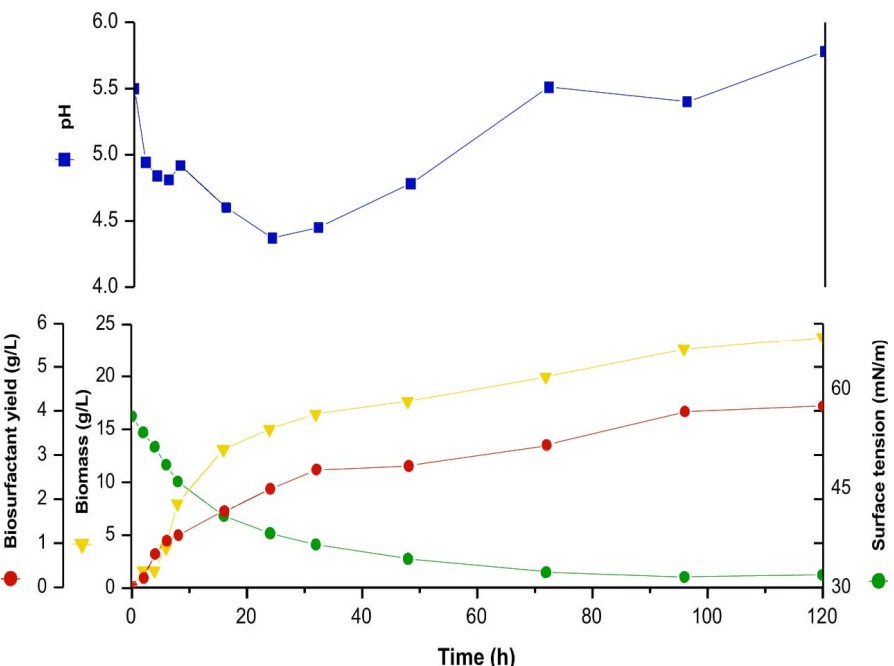

**Figure 1.** Growth, pH, surface tension, and biosurfactant yield curves of *C. tropicalis* UCP 0996 cultivated in distilled water supplemented with 2.5% waste frying oil, 2.5% corn steep liquor, and 2.5% molasses during 120 h at 200 rpm and 28 °C.

*3.2. Stability Studies*

The surface tension of the biosurfactant from *C. tropicalis* UCP 0996 was stable under different NaCl concentrations, various pH, temperature, and heating times at 90 °C (Figure 2A,D). The emulsification capacity of the biosurfactant, on the other hand, showed specific behaviors depending on the environmental condition tested.

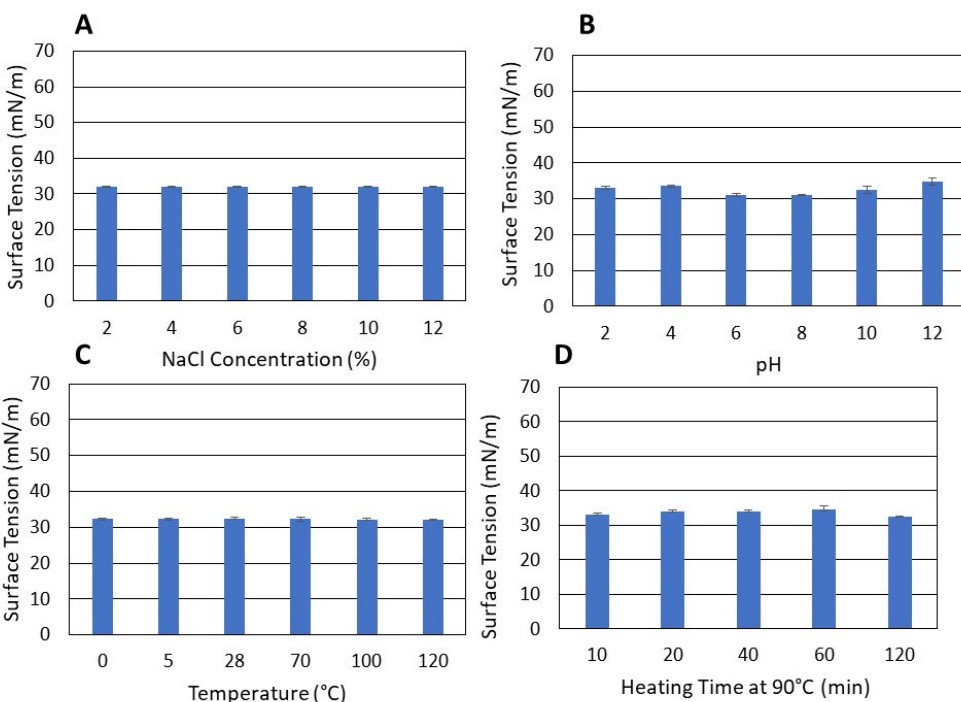

**Figure 2.** Influences of salt concentration (**A**), pH (**B**), temperature (**C**), and heating time at 90 °C (**D**) on surface-tension-reducing activity of cell-free broth containing the biosurfactant from *C. tropicalis* UCP 0996 cultivated in distilled water supplemented with 2.5% cane molasses, 2.5% waste frying oil and 2.5% corn steep liquor for 120 h at 200 rpm and 28 °C.

The emulsification index for the three oils types tested showed to be quite stable, regardless of the salt concentrations tested (Figure 3A). This resistance to salt concentrations shows the potential of the biosurfactant for application in marine environments. In another study, a biosurfactant produced by *C. tropicalis* UCP 1613 [24] also displayed a good performance in different salinity ranges. According to Zuza-Alves et al. [25], *C. tropicalis* yeast has been considered an osmotolerant microorganism and this ability to survive to high salt concentration may be important for fungal persistence in saline environments. This property confirms the *C. tropicalis* potential for biotechnological processes under extreme environmental conditions.

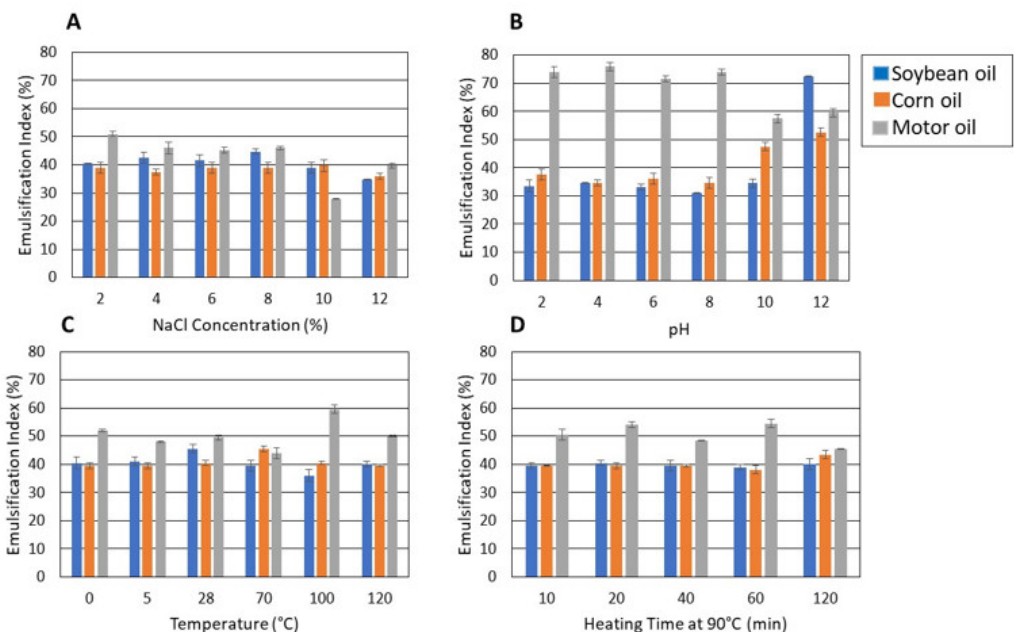

**Figure 3.** Influences of salt concentration (**A**), pH (**B**), temperature (**C**), and heating time at 90 °C (**D**) on the emulsifying index of the cell-free broth containing the biosurfactant from *C. tropicalis* UCP 0996 cultivated in distilled water supplemented with 2.5% cane molasses, 2.5% waste frying oil, and 2.5% corn steep liquor for 120 h at 200 rpm and 28 °C.

The emulsification activities of corn and soybean vegetable oils remained practically stable with pH variations, although an increase between 15 and 35% was observed at pH values of 10–12. Motor oil, on the other hand, showed emulsification rates above 70% at most pH levels, with only a 10% reduction in extreme alkaline values (Figure 3B). According to Camargo et al. [26], the biosurfactants produced by the yeasts *Rhodotorula glutinis* and *Meyerozyma guilliermondii* showed good emulsification performance at acidic pH (2.0), with emulsions around 45.8% and 39.2%, respectively.

The emulsification capacity of the biosurfactant was stable at the investigated temperatures, being maximum at 100 °C for motor oil. The emulsification capacity also remained stable when the biosurfactant was subjected to heating at 90 ° C for 120 minutes (Figure 3C,D). These results show the potential of the biosurfactant for industrial applications at extreme temperatures. According to Luna et al. [23], the surface tension of the biosurfactant from *C. bombicola* remained practically unchanged, demonstrating stability under a wide range of temperature.

### 3.3. Biosurfactant Properties

The isolated biosurfactant from *C. tropicalis* decreased the surface tension of the water from 70.0 to 25.6 mN/m at a concentration of 0.06%, indicating that the CMC was reached in this concentration, as described previously by Almeida et al. [11]. The CMC registered for the biosurfactant from *C. tropicalis* is within the CMC range reported for different types of yeasts biosurfactants, as the biosurfactant from *C. lipolytica* [20], which showed a surface

tension of 28 mN/m and a CMC of 1.4%, and the biosurfactant from *C. glabrata* [27], which showed a surface tension of 28.8 mN/m and a CMC of 2.0%.

Biosurfactants may be located inside the cells (intracellular) or secreted outside the cells (extracellular) [3,28]. The low hydrophobicity found (3.73%) for *C. tropicalis* suggests that under the cultivation conditions applied in this work, the yeast should have used a biosurfactant-mediated alkane uptake as the predominant mechanism, mainly by solubilization of the hydrophobic substrate and formation of micelles with a hydrophilic outer layer. This mode of micellar transfer appears to be well suited to hydrophilic cells, allowing efficient contact with the alkane-degrading micro-organisms [29], while the ionic characterization showed the anionic nature of the biosurfactant.

### 3.4. Biosurfactant Properties

The biosurfactant extracted from the cell-free broth of *C. tropicalis* was analyzed by thin-layer chromatography (TLC). Two spots were produced, which showed positive reactions for sugars and for lipids, but negative reactions for amino groups (Figure 4).

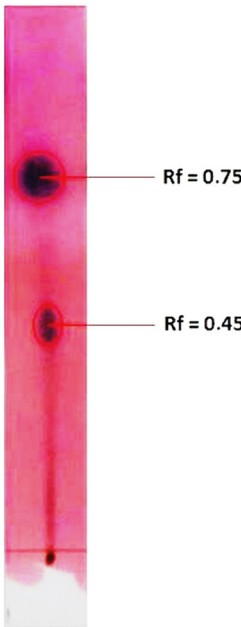

**Figure 4.** TLC of the biosurfactant from *C. tropicalis* UCP 0996 cultivated in distilled water supplemented with 2.5% cane molasses, 2.5% waste frying oil, and 2.5% corn steep liquor for 120 h at 200 rpm and 28 °C.

The FT-IR spectrum of the biosurfactant displayed in Figure 5 shows extending vibration at 3300–3500 cm$^{-1}$, which is characteristic of O–H stretching vibrations, while the band peak at 3000–2800 cm$^{-1}$ is characteristic of aliphatic chains. Vibration around 1710 cm$^{-1}$ evidences the presence of C=O group, and the appearance of band peaks at 1550–1400 cm$^{-1}$ may be due to C double bond, while a band peak at ~1260 cm$^{-1}$ shows the presence of ketone group.

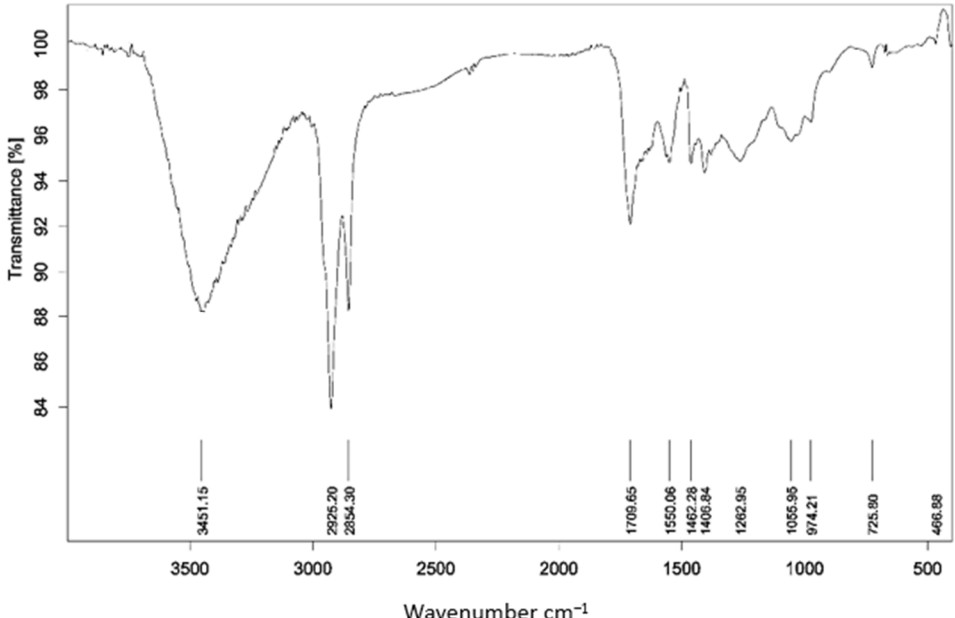

**Figure 5.** FTIR spectrum for biosurfactant extract produced by *C. tropicalis* UCP 0996 cultivated in distilled water supplemented with 2.5% cane molasses, 2.5% waste frying oil, and 2.5% corn steep liquor.

The $^1$H NMR spectrum of the biosurfactant is shown in Figure 6. Signals between δ 0.60 and 1.6 ppm suggested the presence of aliphatic and methyl groups; signals between δ 2.0 and 2.2 ppm indicated the presence of aldehyde group; signals at δ 3.5 ppm and between δ 4.6 and 4.8 ppm were attributed to hydroxyl groups and those between δ 5.0 and 5.4 ppm corresponded to double bounds.

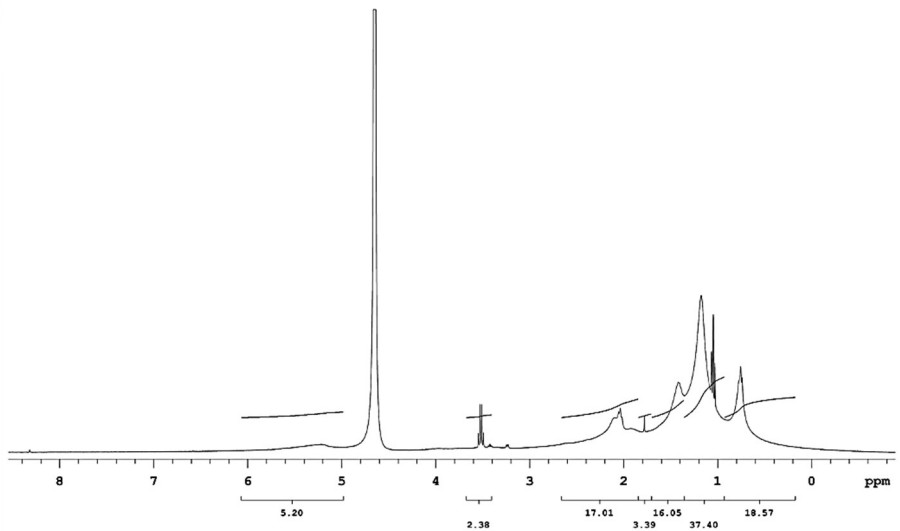

**Figure 6.** $^1$H NMR spectrum (CD$_3$OD, 300 MHz) of the isolated biosurfactant from *C. tropicalis* UCP 0996 cultivated in mineral medium supplemented with 2.5% cane molasses, 2.5% waste frying oil, and 2.5% corn steep liquor.

The biosurfactant was analyzed by GC-MS and compared with the library data. The Chromatogram (Figure 7) showed two very evident peaks probably related to cyclical structures. The first peak (45.53%) is related to a cyclic structure with carbonyl group. The second peak (28.21%) also points to a cyclic structure containing a hydroxyl group. The structure showed molar mass between 150 and 200 ($m/z$).

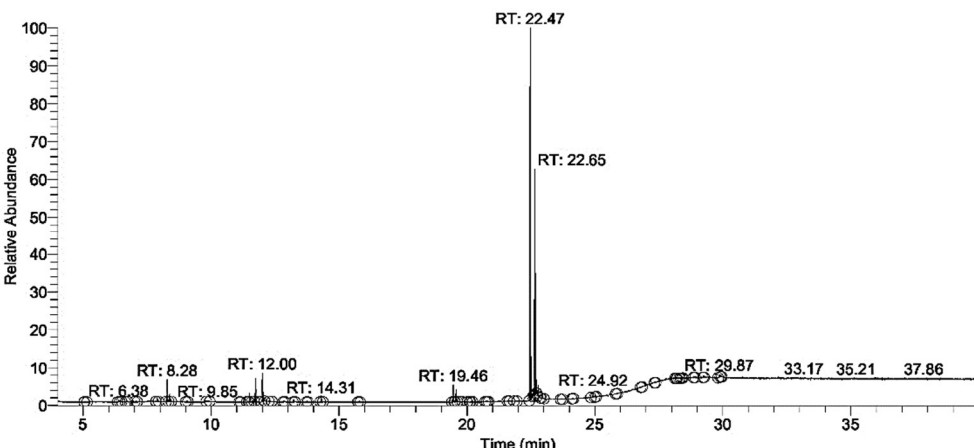

**Figure 7.** Chromatogram of the GC-MS separation of the biosurfactant produced by *C. tropicalis* UCP 0996 showing peaks for cyclic structures containing carbonyl and hydroxyl groups.

The results obtained by [1]H NMR, FTIR spectroscopy, TLC, and GC-MS analysis indicated the glycolipidic nature of the biosurfactant, as it has been described for other yeast biosurfactants in the literature [26,27,30,31].

### 3.5. Toxicity Against Artemia salina as Indicator

The crude and the isolated biosurfactant from *C. tropicalis* was nontoxic to the brine shrimp, since the results indicated that toxicity was only reached at a concentration of 10×CMC (Table 1). Rocha Junior et al. [32] found similar results for the biosurfactant produced by *C. tropicalis* UCP 0996 grown in distilled water with 2.5% molasses, 2.5% frying oil, and 4.0% corn steep liquor, which demonstrated no toxic effects on brine shrimp. Lira et al. [33], using the isolated biosurfactant from *C. guilliermondii* UCP 0992 also showed an *Artemia salina* survival rate of 100%. These findings demonstrate the low toxicity of yeasts biosurfactants.

**Table 1.** Toxicity of the biosurfactant from *C. tropicalis* UCP 0996 cultivated cultivated in mineral medium supplemented with 2.5% cane molasses, 2.5% waste frying oil, and 2.5% corn steep liquor on brine shrimp larvae.

| Concentrations (CMC) | Shrimp Larvae Mortality (%) |
|---|---|
| Cell-free broth | No mortality |
| 1/2×CMC | No mortality |
| 1×CMC | No mortality |
| 2×CMC | No mortality |
| 5×CMC | $10 \pm 0.13$ |
| 10×CMC (LC$_{50}$) | $50 \pm 0.19$ |

### 3.6. Biosurfactant Formulation

A commercial product must have an extended shelf life and during this time; it must be able to maintain its properties over a wide range of pH, temperature, salinity, etc. The stability studies carried out with the *C. tropicalis* biosurfactant demonstrated, in general, that the emulsifying and dispersing properties of the surfactant formulation did not suffer marked changes (Figures 8–10). The surface tension reduction capacity remained practically constant, around 30 mN/m during 120 days of testing by the three preservation methods employed (Figure 8A,H), although the best results were observed with the addition of potassium sorbate to the crude biosurfactant. Regarding emulsfication, the biosurfactant formulated with potassium sorbate promoted a high emulsification rate of motor oil (above 90%) in practically all conditions tested. On the other hand, variations in the emulsification rates were observed with the biosurfactant subjected to fluent vaporization

and fractional tyndallization (Figure 9A,H). The dispersion capacity of motor oil by the biosurfactant, in turn, varied according to the tested environmental condition, although the best results were also obtained after the addition of potassium sorbate, with values above 60% (Figure 10A,H). Based on the results found, the use of potassium sorbate is more advantageous not only from the point of view of efficiency, but also economical, since it eliminates the industrial costs of heat treatment.

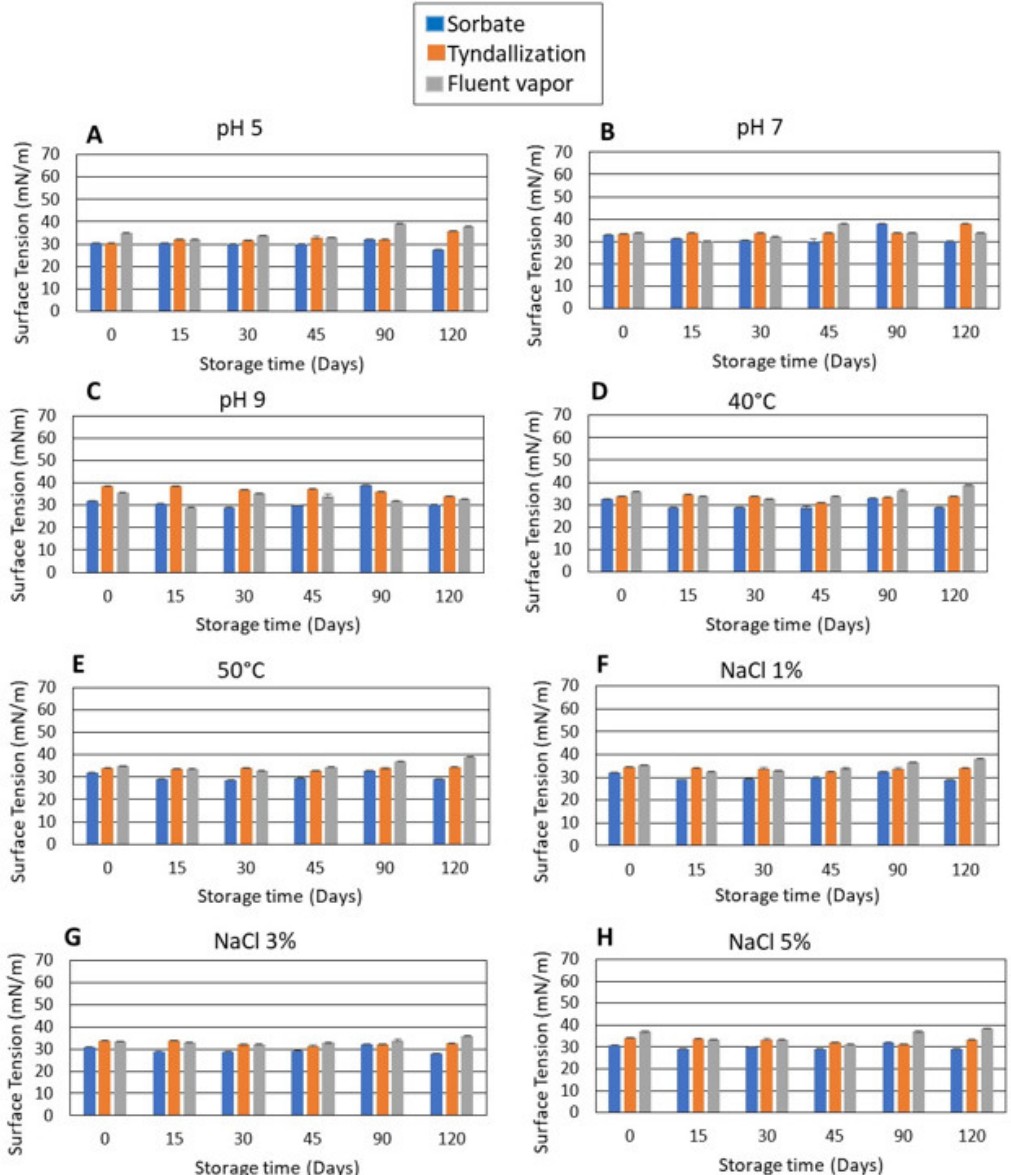

**Figure 8.** Surface tension of the biosurfactant from *C. tropicalis* UCP 0996 formulated with sorbate 0.2% for 120 days under varying pH (**A–C**), temperature (**D–E**), and NaCl (**F–H**) after each storage period.

Freitas et al. [34] formulated a biosurfactant from *C. bombicola* cultivated for application as a dispersant of oil spills. The formulated biosurfactant showed stability through 120 days. Likewise, Santos et al. [35] evaluated the efficiency of a *C. lipolytica* biosurfactant formulated with 0.2% preservative as a bioremediation agent during 120 days. The formulated biosurfactant maintained its activity, demonstrating to be suitable for the formulation of a commercial product. A bacterial biosurfactant from *Bacillus cereus* also maintained its activities during a long period of storage, especially in the presence of the preservative potassium sorbate [36]. Campos et al. [37], on the other hand, tested different mayonnaise formulations with the addition of a bioemulsifier isolated from *C. utilis*. The

most stable formulation was obtained in combination with guar gum and the isolated biosurfactant, which proved to be safe for use in food emulsions. In another study, a rhamnolipid was applied in the formulation of a bio-detergent. The biosurfactant was effective in removing oil from stains and the formulation was comparable to commercial powders in terms of efficiency [38]. Haque et al. [39] evaluated the usefulness of rhamnolipids as non-cytotoxic agents, natural antimicrobials, and antioxidants for various applications as a co-preservative in various product formulations to reduce the use of synthetic preservatives in various industrial and biomedical segments. There are few studies describing the use of biosurfactant formulations in several purposes, which makes this work a more valuable contribution.

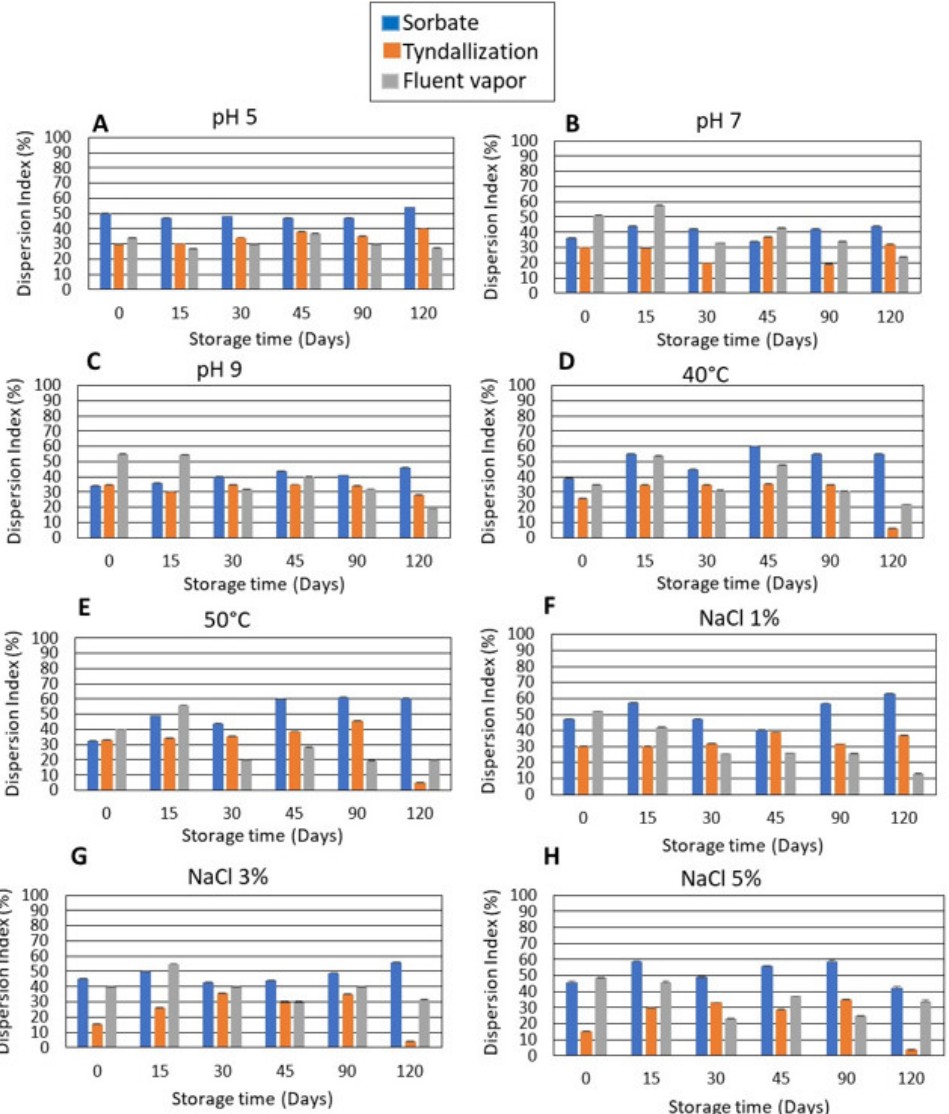

**Figure 9.** Emulsification index of the biosurfactant from *C. tropicalis* UCP 0996 formulated with sorbate 0.2% for 120 days under varying pH (**A–C**), temperature (**D–E**), and NaCl (**F–H**) after each storage period.

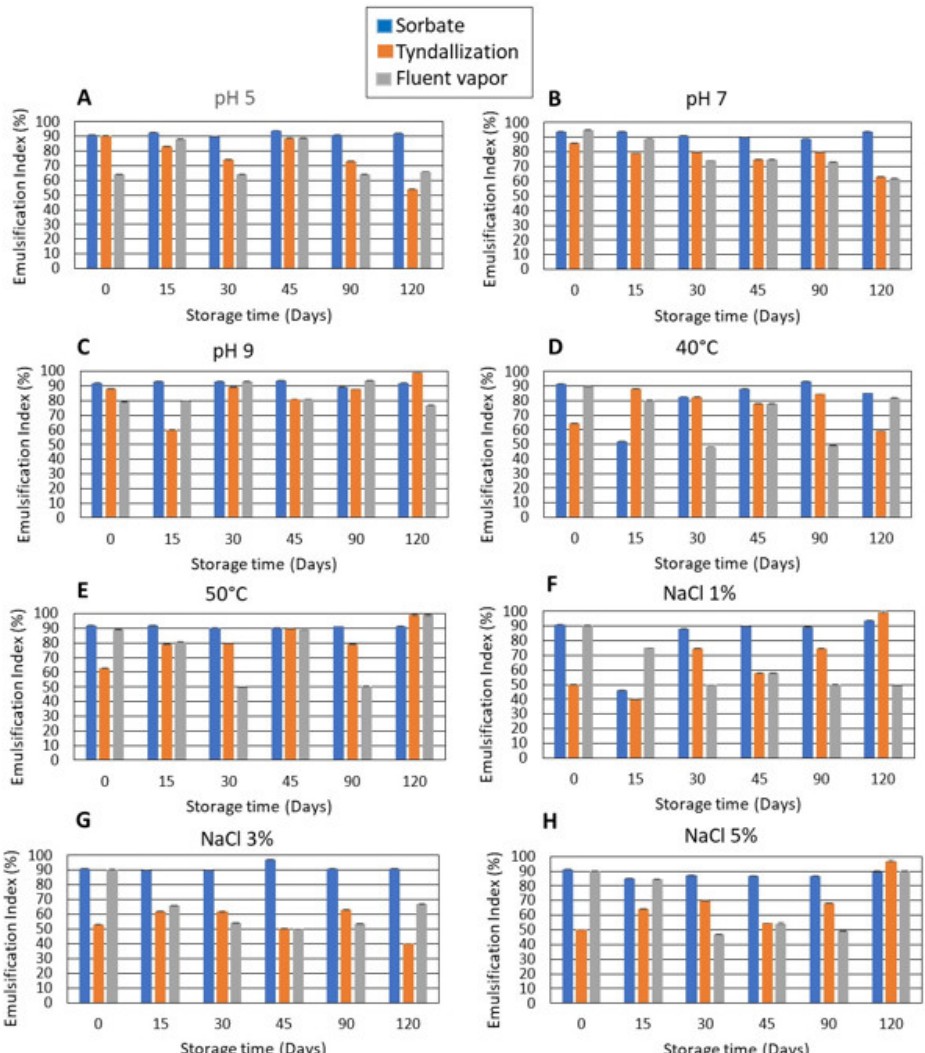

**Figure 10.** Dispersion index of the biosurfactant from *C. tropicalis* UCP 0996 formulated with sorbate 0.2% for 120 days under varying pH (**A**–**C**), temperature (**D**–**E**), and NaCl (**F**–**H**) after each storage period.

### 3.7. *Application of the Biosurfactant in Hydrophobic Contaminant Spreading*

Many processes carried out by the oil industry are responsible for contaminating the marine environment. Eventually, during these events, the oil reaches seawater, requiring the use of different containment technologies, which include the use of surfactants. In this study, the dispersion capacity of a hydrocarbon by the biosurfactant of *C. tropicalis*, which has been previously evaluated by Almeida et al. [11] was subjected to a more detailed evaluation, as shown in Table 2. The higher dispersion rates found were 70.95% and 57.14% for the isolated and formulated biosurfactant, respectively. Another yeast biosurfactant formulated with potassium sorbate also demonstrated a good ability to disperse oil in seawater, with percentages between 30 and 50% after 30 days of storage [32]. Dayamrita et al. [40] evaluated bacteria isolated from oil-contaminated soil as producers of biosurfactants. The supernatant of the cultures showed high surface activity, as they were able to spread oil. In another similar study, Ibrahim [41] demonstrated the dispersion capacity of the biosurfactants produced by bacterial strains isolated from contaminated soil with motor oil. The cell-free broth showed positive results for oil dispersion, proving the presence of biosurfactants.

**Table 2.** Dispersion of motor oil by the biosurfactant from *C. tropicalis* UCP 0996 cultivated in distilled water supplemented with 2.5% waste frying oil, 2.5% corn steep liquor, and 2.5% cane molasses.

| Removal Agent | Dispersion (%) |
|---|---|
| Formulated biosurfactant | $57.14 \pm 0.12$ |
| $1/2 \times$CMC | $38.09 \pm 0.20$ |
| $1 \times$CMC | $45.23 \pm 0.11$ |
| $2 \times$CMC | $53.80 \pm 0.12$ |
| $5 \times$CMC | $70.95 \pm 0.30$ |

*3.8. Washing of Hydrophobic Compound Adsorbed to Porous Surface*

Oil spills in aquatic ecosystems cause irreparable damage to marine life and coastal populations in affected areas [30]. When an oil spill reaches an area of coral reefs, the methods of treatment and recovery become very restricted, as these ecosystems are complex, sensitive, and difficult to access [42]. Among the treatment technologies available, the use of dispersants is one of the most indicated [30]. In this work, the removal of motor oil adsorbed on marine stones by the biosurfactant of *C. tropicalis*, which has been previously evaluated by Almeida et al. [11] was subjected to a more detailed evaluation, as shown in Table 3. The best oil removal rate reached by the biosurfactant from *C. tropicalis* was 66.18%, showing the feasibility of its application as a biological dispersant for removing hydrophobic pollutants in sensitive ecosystems. Few studies describe the action of biosurfactants in cleaning porous surfaces. The biosurfactant of *C. bombicola* removed about 70% of the engine oil adsorbed on a porous surface [23]. In another study, a hydrocarbon removal rate of 84.5% from fragments of coral reefs was reached by a formulated biosurfactant from *Pseudomonas cepacia* [43].

**Table 3.** Removal of motor oil adsorbed on marine stones by the formulated and isolated biosurfactant produced by *C. tropicalis* UCP 0996.

| Removal Agent | Removal (%) |
|---|---|
| Formulated biosurfactant | $41.89 \pm 0.5$ |
| $1/2 \times$CMC | $28.37 \pm 0.3$ |
| $1 \times$CMC | $42.01 \pm 0.12$ |
| $2 \times$CMC | $56.02 \pm 0.21$ |
| $5 \times$CMC | $66.18 \pm 0.4$ |
| Control (distilled water) | $2.35 \pm 0.1$ |

*3.9. Swirling Bottle Test*

The results of the simulation test of the application of the formulated biosurfactant in the dispersion of oil in seawater are shown in Table 4. The best results were observed for the biosurfactant/oil ratio of 1: 1 (*v/v*) for all times tested, with dispersion rates above 50%. The performance of this test demonstrates the ability of the biosurfactant to reduce the effects of large oil spills, since the dispersion of the oil in small droplets increases the exposure surface of the contaminant, facilitating its degradation by indigenous microorganisms [3,19,44]. Soares da Silva et al. [45] used the Swirling Bottle technique to apply a formulated and isolated microbial surfactant in an oily effluent.

**Table 4.** Evaluation of the biosurfactant from *C. tropicalis* UCP 0996 cultivated in distilled water supplemented with 2.5% waste frying oil, 2.5% corn steep liquor, and 2.5% cane molasses as an oil spill dispersant.

| Biosurfactant/Oil Ratio (*v/v*) | Resting Time (min) | Dispersion (%) |
|---|---|---|
| 1/1 | 0 | $65.03 \pm 0.50$ |
| | 5 | $59.45 \pm 0.10$ |
| | 10 | $50.23 \pm 0.30$ |
| 1/2 | 0 | $41.13 \pm 0.11$ |
| | 5 | $31.50 \pm 0.60$ |
| | 10 | $27.11 \pm 0.23$ |
| 1/8 | 0 | $20.47 \pm 0.20$ |
| | 5 | $17.31 \pm 0.14$ |
| | 10 | $14.34 \pm 0.16$ |
| 1/20 | 0 | $11.26 \pm 0.50$ |
| | 5 | $7.92 \pm 0.20$ |
| | 10 | $4.15 \pm 0.10$ |

*3.10. Bioremediation Test*

The results of motor oil bioremediation through stimulation of degradation activity of indigenous marine bacteria and fungi by the presence of the biosurfactant was previously described by Almeida et al. [11] for the isolated biosurfactant from *C. tropicalis*. In this work, the same experiment was carried out including the formulated biosurfactant and other isolated biosurfactant concentration during 28 days of experiment (Figure 11). Experiments carried out in the presence of the biosurfactant were superior in terms of microorganism's growth and hydrocarbon degradation compared to the control (in the absence of the biosurfactant). This result suggests that both the formulated and isolated biosurfactants from *C. tropicalis* had ability as a bioremediation additive. Santos et al. [35] obtained similar results for the biosurfactant from *C. lipolytica*, which stimulated the growth of indigenous microorganisms in seawater during 30 days of cultivation.

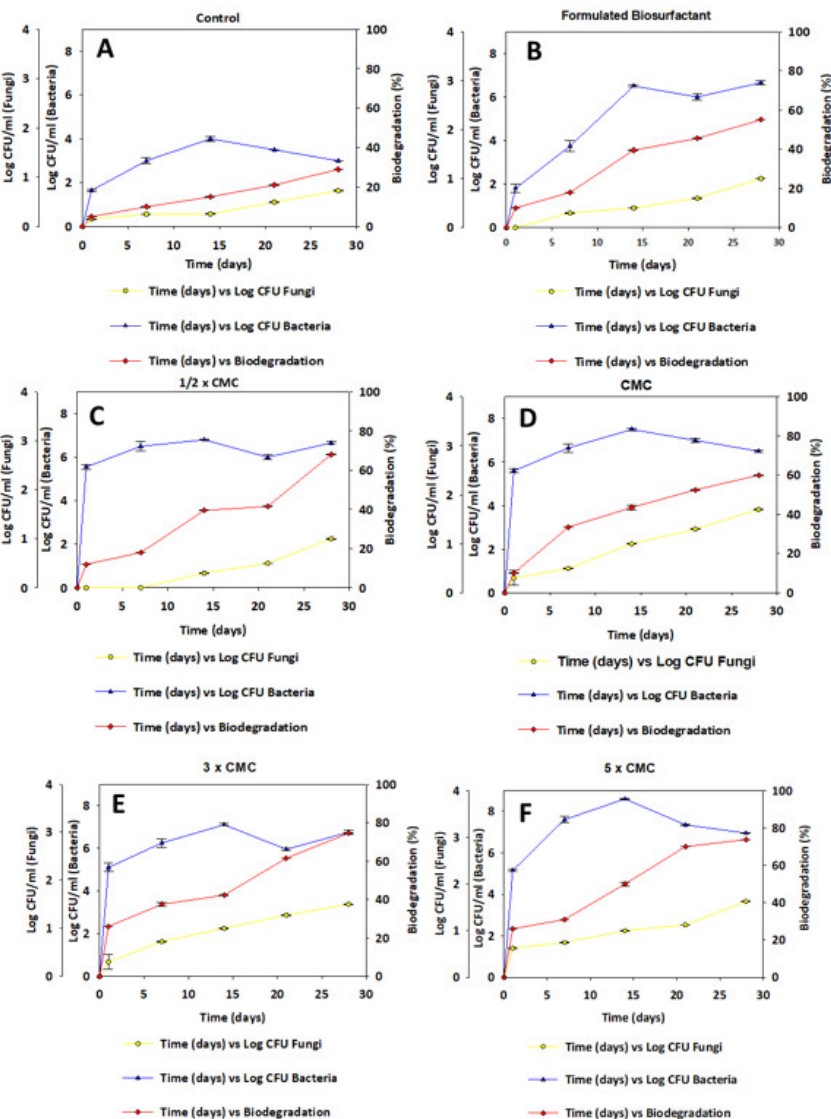

**Figure 11.** Influence of the biosurfactant from *C. tropicalis* UCP 0996 on growth of bacteria and fungi indigenous in seawater. Microbial growth in the absence of biosurfactant (**A**) and in the presence of the formulated biosurfactant (**B**) and in the presence of the isolated biosurfactant at $\frac{1}{2}\times$CMC (**C**); at the CMC (**D**); at 3×CMC (**E**); and at 5×CMC (**F**).

## 4. Conclusions

The biosurfactant produced by *C. tropicalis* grown in a low cost medium has considerably reduced the surface tension of water, demonstrating a high emulsifying and dispersing capacity of hydrophobic compounds. The biosurfactant also demonstrated excellent stability in extreme environmental conditions of salinity, temperature, and pH variations, in addition to not presenting toxicity to marine organisms under the conditions tested. The formulated biomolecule showed an extended shelf life, showing considerable potential for application as a commercially stable dispersant, revealing the viability of applying the biomolecule without the need for product purification.

**Author Contributions:** Conceptualization, L.A.S.; methodology, D.G.A., R.d.C.F.S.d.S., and H.M.M.; validation, E.J.S. and P.P.F.B.; formal analysis, D.G.A., P.P.F.B. and L.A.S.; investigation, D.G.A., E.J.S., H.M.M. and R.d.C.F.S.d.S.; resources, L.A.S.; data curation, D.G.A., J.M.L., and R.D.R.; writing—original draft preparation, D.G.A. and L.A.S.; writing—review and editing, L.A.S.; visualization, L.A.S.; supervision, L.A.S.; project administration, L.A.S.; funding, L.A.S.; acquisition, L.A.S. All authors have read and agreed to the published version of the manuscript.

**Funding:** This study was funded by the Programa de Pesquisa e Desenvolvimento da Agência Nacional de Energia Elétrica (ANEEL)/Thermoelectric EPASA (Centrais Elétricas da Paraíba)/ Thermoelectric EPESA (Centrais Elétricas de Pernambuco S.A.)/Termocabo S.A. (Grant n. PD-07236-0009/2020) and by the Brazilian development agencies Fundação de Apoio à Ciência e Tecnologia do Estado de Pernambuco (FACEPE), Conselho Nacional de Desenvolvimento Científico e Tecnológico (CNPq) and Coordenação de Aperfeiçoamento de Pessoal de Nível Superior (CAPES) (Grant n. Finance Code 001).

**Institutional Review Board Statement:** Not applicable.

**Informed Consent Statement:** Not applicable.

**Data Availability Statement:** Not applicable.

**Acknowledgments:** The authors are grateful to the Laboratories from the Universidade Católica de Pernambuco (UNICAP) and from the Instituto Avançado de Tecnologia e Inovação (IATI), Brasil.

**Conflicts of Interest:** The authors declare no conflict of interest.

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
