# Peer review of "Production, Characterization and Commercial Formulation of a Biosurfactant from Candida tropicalis UCP0996 and Its Application in Decontamination of Petroleum Pollutants"

_processes, doi:10.3390/pr9050885_

Round 1

Reviewer 1 Report

Thank you for the article. This is very good research. The topic is actual and the results are useful for the applied research. The results of the analysis are based on a whole range of tests. The methods and statistical analysis are chosen properly. I do not have critical comments.

Author Response

Dear reviewer, many thanks for the positive comments made about our manuscript. We are very grateful for your evaluation.

Reviewer 2 Report

The work entitled “Production, characterization and commercial formulation of a biosurfactant from Candida tropicalis UCP0996 and its application in decontamination of petroleum pollutants”, authors are D.G. Almeida, R.C.F. Soares da Silva, H.M. Meira, P.P.F. Brasileiro, E.J. Silva, J.M. Luna, R.D. Rufino, and L.A. Sarubbo, is devoted to characterization of the biosurfactant produced by yeast Candida tropicalis UCP0996. This biosurfactant could be used potentially as bioremediation agent for removal of hydrocarbon pollutants and as food preservative. Search for novel, safe and effective surface-active agents is very actual in terms of replacement of widely used synthetical surfactants being a widely met contaminant. Various experiments providing full description of functional characteristics of the UCP0996 biosurfactant were done by authors, including information about the biosurfactant activity, stability and chemical composition. Different methods were used, such as tensiometry, MATH, thin chromatography, NMR spectroscopy, Fourier Transform Infrared Spectroscopy, GC-MS, and toxicity test.

Major revision

Some results given in the paper are published already in the article Almeida et al., Chem Eng Transactions, 2018, 64, 541–546, doi 10.3303/CET1864091. These results include determination of critical micelle concentration (Fig. 4 in this study and Fig. 1 in the Almeida article), motor oil dispersion and removal (Tables 2 and 3 in this study and Tables 1 and 2 in the Almeida article), and influence of the biosurfactant on growth of bacteria and fungi indigenous in seawater (Fig. 12 in this study and Fig. 2 in the Almeida article). Duplicated results should be removed from the paper. If it is necessary, authors can refer to the Almeida article in Introduction or somewhere in Results.

Minor revisions

Figures 2,3,9,10,11,12 – Lines for errors (standard deviations) are very tiny and poorly visible. Also titles for axes are diffused and not readable.

p. 3, line 122 – “one hundred ml of n-hexadecane”? It seems to be an error, 100 ml does not fit into 2 ml tube…

p. 6, line 241 – “Growth”, not “groth”.

p. 6, line 270 – “C. tropicalis”, not “C. Tropicalis”.

p. 8, lines 316-318 – Could authors explain how low hydrophobicity revealed that the biosurfactant was not attached to the membrane?

Summary. Overall recommendation: Reconsider after major revision.

Round 2

Reviewer 2 Report

Authors have taken into consideration all comments and improved the manuscript. The work entitled “Production, characterization and commercial formulation of a biosurfactant from Candida tropicalis UCP0996 and its application in decontamination of petroleum pollutants”, authors are D.G. Almeida, R.C.F. Soares da Silva, H.M. Meira, P.P.F. Brasileiro, E.J. Silva, J.M. Luna, R.D. Rufino, and L.A. Sarubbo, can be published in its current form.